# A Self-Supervised Framework for Function Learning and Extrapolation

**Simon N. Segert**                                                      *ssegert@princeton.edu*
*Princeton Neuroscience Institute*
*Princeton University*

**Jonathan D. Cohen**                                                      *jdc@princeton.edu*
*Princeton Neuroscience Insitute*
*Princeton University*

**Reviewed on OpenReview:** `https://openreview.net/forum?id=ILPFasEaHA`

## Abstract

Understanding how agents learn to generalize — and, in particular, to extrapolate — in high-dimensional, naturalistic environments remains a challenge for both machine learning and the study of biological agents. One approach to this has been the use of function learning paradigms, which allow agents' empirical patterns of generalization for smooth scalar functions to be described precisely. However, to date, such work has not succeeded in identifying mechanisms that acquire the kinds of general purpose representations over which function learning can operate to exhibit the patterns of generalization observed in human empirical studies. Here, we present a framework for how a learner may acquire such representations, that then support generalization — and extrapolation in particular — in a few-shot fashion in the domain of scalar function learning. Taking inspiration from a classic theory of visual processing, we construct a self-supervised encoder that implements the basic inductive bias of invariance under topological distortions. We show the resulting representations outperform those from other models for unsupervised time series learning in several downstream function learning tasks, including extrapolation.

## 1 Introduction

A key feature of an intelligent agent is the ability to recognize and extrapolate a variety of abstract patterns that commonly occur in the world. Here, we focus on a tractable but still highly general special case of such patterns, that take the form of one-dimensional smooth functions. From a formal perspective, the space of all such functions is vast (Reed and Simon, 1980), necessitating the use of inductive biases for making useful inferences. Thus, while this setting does not encompass all possible kinds of structures that can be generalized or extrapolated, it is a sufficiently rich space that insights gained here are likely to shed light on the ability of biological and artificial agents to generalize more broadly. At the same time, the abstract structure of this space is relatively simple and well-understood, thus making it amenable to precise analysis and interpretable experimental manipulations.

A further virtue of the space of functions is the existence of detailed experimental data from humans in this domain, thus facilitating a direct comparison of models with natural agents. Indeed, over the past few decades, the empirical studies of function learning in humans has catalogued the forms of several such commonly applied biases, including associative similarity, rule based categorization (McDaniel and Busemeyer, 2005), bias towards positive linear forms (Kwantes and Neal, 2006) and compositional construction from small number of basis elements (Schulz et al., 2017). Taken together, such results describe a class of "intuitive functions," which are functions that people appear readily able to recognize and use.

While it is generally accepted that efficient generalization implies the existence of some previous expectations about the structure of the space of functions, what is not obvious is how such expectations are acquired; that is, what mechanisms are capable of learning and encoding abstract structure through unsupervised or self-supervised experience, in such a way that features relevant to any particular task may be easily "read out" as required. Here, we propose to address these challenges by adapting and extending the general framework of the field of self-supervised learning (Chen et al., 2020; He et al., 2020). The framework consists of two components: a "slow" encoder that learns general purpose representations of one-dimensional functions using a standard self-supervised learning algorithm, and a collection of "fast" heads, which can rapidly adapt to different function learning paradigms, based on a small amount of task-specific annotated data, using a simple form of supervised learning (linear or logistic regression). The heads are trained on top of the representations learned previously by the encoder, allowing the model to make use of its general knowledge to rapidly adapt to the particular task demands.

While efforts have taken this general approach (Chen et al., 2020; He et al., 2020), none to our knowledge have specifically considered the domain of function learning with intuitive functions – that is, ones that people have been empirically observed to use (DeLosh et al., 1997; McDaniel and Busemeyer, 2005; Schulz et al., 2017). Our approach is further distinguished in the design of the encoder used for self-supervised learning. For this, we treat a scalar function as a (typically very short) time series. The crucial feature of our encoder is a novel family of augmentations of time series, derived from the theory and phenomenology of topological visual processing (Zeeman, 1965; Chen, 2005). This theory holds that the visual system is invariant to certain kinds of local topological distortions of stimuli, distortions that we design our augmentations to mimic. We hypothesize that such distortions reflect commonly occurring structure in the world, that may in turn have been discovered by the brain, either through evolution or early development, and used as a basis for generalization. Following this idea, we train on a self-supervised objective that tries to enforce invariance across these augmentations, adapting the framework of Chen et al. (2020).

We demonstrate that our choice of encoder and training procedure learns representations that perform better on a collection of downstream function learning and generalizaton tasks than do comparison models for learning and/or representing time series. This should be of particular interest to the field of semi-supervised learning, since works in that field have not yet systematically analyzed time series that correspond to intuitive functions. Moreover, we directly compare the generalization patterns of the model with those of humans asked to perform a multiple-choice extrapolation paradigm modeled after an empirical study by Schulz et al. (2017). We find that the model exhibits a qualitatively similar bias as people in this setting, namely, a greater accuracy on functions that are compositionally structured. This should also be of interest to psychologists, since it suggests that behavioral biases in function learning may arise as consequences of a more general representation-learning procedure.

## 2 Background

### 2.1 Contrastive Learning

Here we provide a brief summary of the elements of contrastive learning that are necessary to define our encoder. This is adapted from Chen et al. (2020) and van den Oord et al. (2018). The basic assumption is that we are provided with a set of positive pairs $(v_i, v_i'), i = 1, \ldots, N$, which are taken as inputs that we wish to consider similar to each other. All other pairs of inputs are considered as negative pairs, which the objective will attempt to push apart in the latent space. For convenience, we will treat the input as a single flattened dataset of size $2N$, in which the positive pairs are those of the form $(v_i, v_{i+N}), i \leq N$. Let $f_\theta : \mathbb{R}^{n_1} \to \mathbb{R}^{n_2}$ and $g_\phi : \mathbb{R}^{n_2} \to S^{n_3-1}$ be two parametric families of functions (e.g. neural networks). Here $n_1$ is the dimensionality of the inputs $v_i$, while $n_2$ and $n_3$ are arbitrary, and $S^{n_3-1}$ denotes the hypersphere consisting of all vectors in $\mathbb{R}^{n_3}$ of unit norm. The objective is

$$\max_{\theta, \phi} \sum_{i=1}^{2N} < (g_\phi \circ f_\theta)(v_i), (g_\phi \circ f_\theta)(v_{i+N}) > -LSE_{j \neq i}^\tau (< (g_\phi \circ f_\theta)(v_i), (g_\phi \circ f_\theta)(v_j) >) \tag{1}$$

Here, $\tau > 0$ is a hyperparameter, and $LSE$ denotes the logsumexp function $LSE_i^\tau(z_i) := \tau * \log \sum_i e^{z_i/\tau}$. The brackets $< \cdot, \cdot >$ are the Euclidean dot product. After optimizing this objective, we discard the function $g$ and take the encoder to be the function $f$.

Informally, the first term of the objective function acts to push positive pairs together in the latent space, since it is maximized when both elements in the pair have equal representations. Conversely, the second term acts to push apart all other pairs of inputs. This is because the logsumexp is a monotonically increasing function of each of its inputs; therefore it will be minimized when all of the pairwise similarities are as small as possible. A more precise analysis of properties of this objective may be found in Wang and Isola (2020).

### 2.2 A Generative model of Intuitive Functions

To define a generative model for reference curves that plausibly resemble the distribution encountered by people, we adapt the generative process proposed in Schulz et al. (2017). This generative model uses the formalism of Gaussian processes Rasmussen and Williams (2006); we provide further general background in the Appendix.

Schulz et al. (2017) define the Compositional Grammar by starting from three basic Gaussian Process kernels:

$$
\begin{aligned}
K_{linear}(x,y) &= (x - \theta_1)(y - \theta_1) \\
K_{rbf}(x,y) &= \theta_3 e^{-(x-y)^2/\theta_2^2} \\
K_{periodic}(x,y) &= \theta_4 e^{-\sin^2(2\pi|x-y|\theta_5)/\theta_6^2}
\end{aligned}
$$

where $\theta_i$ are hyperparameters. In addition, the authors include in the CG ten kernels which are defined using pointwise sums and products of these above three. We refer to the Appendix for a more detailed description.

A natural point of comparison is the Spectral Mixture (SM) kernel (Wilson and Adams, 2013), which is a flexible non-parametric kernel family defined by the formula

$$
K_{mix}(x,y) = \sum_{i=1}^{m} w_i e^{-2\pi^2(x-y)^2\sigma_i} \cos(2\pi(x-y)\mu_i)
$$

Schulz et al. (2017) demonstrated that people learn curves generated from the CG more easily than ones generated from the SM . Therefore the family of kernels in the CG are good candidates for generating curves that are both naturalistic and are easily recognized by people.

Lastly, it is important to note that, due to the nature of continuous space, in practice it is necessary to represent functions by their values on a finite set $x_1 < \ldots < x_N$ of ordered sample points. In our case, we take the points to be evenly-spaced, and use the same set of points for every function. Thus any function $\{(x_i, y_i)\}$ may be treated as a time series and vice versa[1]. In what follows we will use the terms "curve," "function" and "time series" interchangeably, with the understanding that the points $x_i$ remain fixed across all functions. Also, since the positions of the $x_i$'s are the same for all functions, we omit them from explicit notation, and use $y$ to denote the vector with components $\{y_i\}_i$ that defines a function.

## 3 A Contrastive Encoder for Intuitive Functions

To define the encoder, following Section 2.1, we need to specify the architecture and the family of positive pairs. For the encoder architecture, we simply take $f$ to be a feedforward network of several 1D convolutions, and $g$ to be an MLP with a single hidden layer. We set $n_2 = n_3 = 128$ and $\tau = .5$. For the class of augmentations, we take inspiration from the field of topological visual perception (Chen, 2005; Zeeman, 1965), which posits that the visual system maintains an invariance to local topological distortions (or "tolerances") of stimuli in order to facilitate global processing.

---

[1]In function learning, the x-axis does not necessarily correspond to time. The same is true of a "time series," despite the name: it is merely an ordered list of numbers.

In our case, we consider 1-dimensional functions rather than 2-dimensional images, but similar principles apply. We propose a family of transformations that implements localized topological distortions to the function, together with several basic global distortions: (1) random vertical reflection, (2) random jittered upsampling and (3) random rescaling. We denote these respectively by stochastic transformations $T_1, T_2, T_3$, which we describe in more detail below.

The first transformation, that accommodates vertical reflection, is defined by $T_1(y) = -y$ with 50 percent probability and $T_1(y) = y$ otherwise. The second, that accommodates horizontal bending, is the most elaborate. To evaluate $T_2(y)$, we first select a random interval $[a, b] \supset [x_1, x_T]$. We then randomly select points $x'_1 < \ldots < x'_T$ in $[a, b]$. These points are not required to be uniformly spaced. We generate them by sampling uniformly and independently at random from $[a, b]$ and then sorting the samples, and then define $T_2(y)_i = \sum_j C_i e^{-\frac{(x'_j - x_i)^2}{2\sigma^2}} y_j$ where $1/C_i = \sum_j e^{-\frac{(x'_j - x_i)^2}{2\sigma^2}}$. In other words, the values of $T_2(y)$ are given by a Gaussian Kernel Density Estimator (KDE) at the points $x_i$. The effect is three-fold: since the points $x_i$ lie in a proper sub-interval of $[a, b]$, this crops a portion of the function and up-samples to the original resolution. Secondly, because the points $x'_i$ are not evenly spaced, some inhomogeneous horizontal stretching or contraction is introduced. Thirdly, the nature of the Gaussian KDE means that the augmented functions are smoothed with respect to the originals. Finally, we apply $T_3$ that accommodates vertical rescaling. For this, we choose a random interval $[a, b] \subset [0, 1]$ and then apply an affine transformation such that the maximum value of the new function is $b$ and the minimum is $a$. More explicitly, $T_3(y)_i = (b - a)\frac{y_i - \min_j y_j}{\max_j y_j - \min_j y_j} + a$. Since this is applied last, the resulting functions always take values in the interval $[0, 1]$. Therefore the positive pairs take the form $(T_3 T_2 T_1(y), T_3 T_2 T_1(y))$ where $y$ is a function. We reiterate that $T_i$ are stochastic transformations, so despite the notational appearance, the two functions comprising a given positive pair will not be equal, since they are generated using two separate evaluations of a stochastic transformation. In our experiments the function $y$ itself is generated from the Compositional Grammar over intuitive functions described in Section 2.2. An illustration of the augmentations is provided in Figure 1.Furthermore, we provide ablation studies on the effect of each augmentation individually in the Appendix.

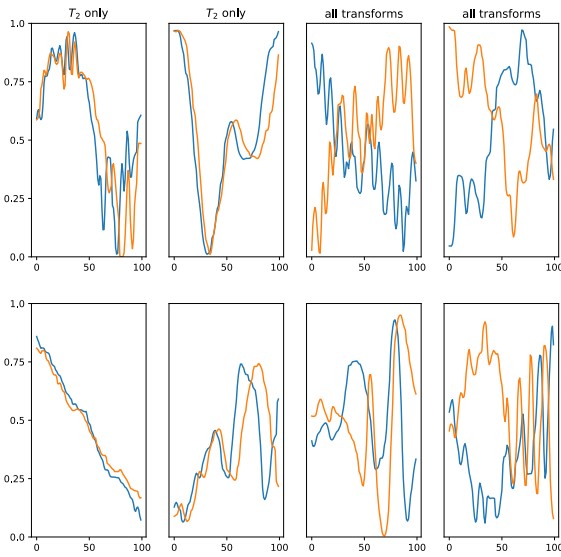

Figure 1: Illustrations of the augmentations. Each plot consists of two functions comprising a positive pair. In the four plots on the left, only the horizontal stretch transformation $T_2$ is applied. In the four plots on the right, all three transformations are applied.

## 4 Data description and Encoder training

To evaluate the ability of the encoder to learn a representation of intuitive functions, we generated and trained it on two types of functions: one generated from the family of 13 kernels defined by the CG (see Section 2.2); and the other (used as a control) from a non-compositional SM Kernel, for a total of 14 kernels. As noted above, Schulz et al. (2017) showed that human completions are closer to those generated by the CG than by the SM. We included the SM in our generative distribution to allow a similar comparison. Each function was generated by first sampling one of these 14 kernels, then sampling any hyperparameters of that kernel, and finally sampling from the resulting covariance matrix evaluated on $T = 100$ evenly horizontally spaced points. We sampled from the SM kernel 50 percent of the time, and each of the remaining 13 CG kernels $3.85(= 50/13)$ percent of the time. Therefore, any differences in the representations between the SM and the CG cannot be ascribed to data availability. We normalized all functions to lie in the interval $[0, 1]$. All encoders were trained using a batch size of 512, with an Adam optimizer with learning rate of .001 and weight decay of $10^{-6}$. All encoders were exposed to 500,000 curves during training. We trained three copies of each encoder using random initializations and averaged results over these copies.

As described above, our model consists of a combination of a contrastive loss function, and a convolutional encoder architecture. In addition, we considered eight comparison models, six of which are encoding models trained using different objectives than the contrastive loss, and two of which are architectural ablations trained using the same contrastive loss, but with non-convolutional encoder architectures. Of the first six, four were unsupervised time series models: Triplet Loss (tloss) (Jean-Yves et al., 2019), Temporal Neighborhood coding (tnc) (Tonekaboni et al., 2021), Contrastive Predictive Coding (cpc) (van den Oord et al., 2018), and Conditional Neural Processes (cnp) (Garnelo et al., 2018a). We also tested a Variational Autoencoder (vae) (Kingma and Welling, 2014) as an example of an unsupervised algorithm that has been successful in other domains, but does not exploit any structure particular to time series. Finally, we included a baseline encoder ("raw") that simply copies the raw input. To control for the latent space capacity, all encoders except for the baseline had representations of equal dimensionality (128).

The first of the two architectural ablations consisted of replacing the convolutional encoder with a multi-layer perceptron. We denote this by "contrastive-mlp." In the second, we replaced the convolutional encoder with the same permutation-invariant encoder as was used in the CNP model, which we denote by "contrastive-perm-inv". In this encoder, the representation of the function $\{(x_i, y_i)\}_{i=1}^{n}$ takes the form $\frac{1}{n} \sum_{i=1}^{n} MLP(x_i, y_i)$ (Garnelo et al., 2018a;b). This encoder is thus permutation invariant in the sense that the representation of the function does not depend on the ordering of its constituent x-y observations. For both of these ablations, the architectures were constrained to have approximately the same number of parameters as the convolutional encoder. Further implementational details of all comparison models are provided in the Appendix.

## 5 Results on Downstream Classification and Extrapolation tasks

To evaluate the quality of the learned representations, we adapted three function learning paradigms that are either directly translated from or inspired by paradigms from studies of human performance: (1) kernel classification, (2) multiple choice extrapolation, and (3) freeform extrapolation. The first one corresponds directly to the standard paradigm for unsupervised learning evaluation in computer vision (Chen et al., 2020), in which an unsupervised algorithm is trained on a dataset for which ground truth annotations are available, and then a supervised classifier such as a logistic regression is fit on top of the frozen representations. Although to our knowledge this has not been used directly in the analysis of empirical results concerning human function learning, it may be regarded as an abstraction of the experiments from Leon-Villagra and Lucas (2019), which showed that people's completions depend on their judgements about the category to which a function belongs, suggesting that people make use of categories when judging functions. The two extrapolation tasks are drawn directly from Schulz et al. (2017). A version of the third task also appears in Wilson et al. (2015), however using a different generative process for the probe curves.

For each task, we define a head that transforms the encoder representations to a task-specific output, and train the head on a small amount of labeled data. In all cases when training the heads, the weights of the encoder are frozen.

Table 1: Accuracy on the categorization task, as a function of the number of training examples per category. Chance performance is 7.14 percent

|  | 3 | 10 | 30 | 100 | 300 |
|---|---|---|---|---|---|
| contrastive | **40.95± 1.83** | **55.27± 1.41** | **64.81± 1.80** | **72.00± 1.52** | **76.23± 1.31** |
| cnp | 15.07± 1.28 | 19.25± 1.35 | 22.69± 0.96 | 26.19± 0.95 | 28.10± 0.73 |
| cpc | 25.06± 1.29 | 35.45± 1.57 | 46.38± 1.14 | 54.94± 1.18 | 58.48± 1.04 |
| raw | 11.86± 1.47 | 15.54± 2.02 | 14.27± 1.96 | 15.73± 1.51 | 14.25± 1.64 |
| t-loss | 30.41± 1.73 | 41.31± 1.89 | 52.16± 1.20 | 59.78± 1.24 | 63.35± 1.19 |
| tnc | 23.15± 1.14 | 31.22± 1.09 | 38.55± 1.23 | 44.85± 1.08 | 49.16± 1.20 |
| vae | 9.27± 1.13 | 12.77± 1.65 | 21.51± 1.89 | 29.17± 1.56 | 33.74± 1.50 |
| contrastive-perm-inv | 16.24± 1.48 | 22.89± 1.43 | 27.09± 0.88 | 29.37± 0.70 | 31.75± 0.87 |
| contrastive-mlp | 33.58± 2.05 | 46.08± 1.37 | 53.96± 0.90 | 57.79± 0.67 | 60.30± 0.65 |

In addition, we note that several of the tasks required that we compute the posterior mean with respect to a Gaussian kernel in order to construct the training data, which required that we initially fit the kernel hyperparameters. For example, in the multiple choice task, to construct two candidate completions we took the posterior mean of the prompt curve with respect to both the SM and the CG kernels, which required we first fit the hyperparameters for those kernels. The fitting of GP kernel hyperparameters is known to suffer from under-fitting and instability problems (see Wilson et al. (2015), including the supplementary material). To address this, we fixed the hyperparameter values of each kernel class, and evaluated the downstream performance of all encoders on curves generated with this fixed set of hyperparameters. We repeated this 10 times using different random choices of hyperparameters each time, and averaged the results. This ensured that the hyperparameter values were correctly specified within each task, and that our results were not influenced by the imperfections of any particular hyperparameter optimization procedure.

We trained three copies of each of the six encoders (contrastive encoder and five comparisons) using different initializations, and all reported results were averaged over the 3 copies and 10 hyperparameter choices, for a total of 30 measurements. The error bars are 95 percent confidence intervals of the standard error of the mean over those measurements.

## 5.1 Kernel classification

Here, the task was to predict which of the 14 kernels was used to generate a given function. The head was simply a linear layer + softmax, the outputs of which were interpreted as the probabilities of each class. Thus it is equivalent to a 14-way logistic regression on the encoder representations. In all cases, we fit the head using the SGDClassifier class from scikit-learn. Additionally, we separately chose an L2 penalty for each head using cross validation. We report the accuracy of each such classifier on a collection of 2800 held-out curves (200 per class). As shown in Table 1, the contrastive encoder is able to attain approximately 55 percent accuracy using only 10 labeled examples per class, which improves to approximately 75 percent when using 300 examples per class, improving upon the second-best model (t-loss) by around 10 percentage points.

## 5.2 Multiple Choice Extrapolation

In the multiple choice completion paradigm, the models were presented with a prompt curve $y \in \mathbb{R}^{80}$, as well as several candidate completions curves $y^i \in \mathbb{R}^{100}$ with the properties that $y^i_j = y_j$ for $j <= 80$ and were required to select the correct completion. Following Schulz et. al., we constructed the candidate completion curves by computing the posterior mean with respect either to the SM kernel, or to the best-fitting CG kernel, with the correct answer corresponding to which of these two kernels was used to generate the prompt curve.

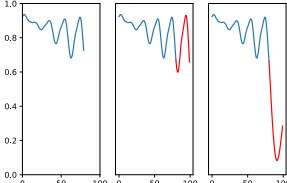

Figure 2: An example multiple choice completion problem. The prompt curve is on the left. The compositional completion is in the middle and the mixture completion is on the right. In this case, the correct answer is the compositional completion. The coloring of the candidate curves is for visual aid only.

Table 2: Performance on the Multiple Choice completion task, as a function of the number of training examples per category. Chance performance is 50 percent.

|                      | 3            | 10           | 30           | 100          | 300          |
| -------------------- | ------------ | ------------ | ------------ | ------------ | ------------ |
| contrastive          | **67.74± 2.48** | **73.68± 1.69** | **78.09± 1.47** | **79.07± 1.75** | **80.90± 1.93** |
| cnp                  | 59.30± 2.35  | 59.80± 2.35  | 59.72± 2.10  | 60.04± 2.14  | 61.56± 2.30  |
| cpc                  | 62.80± 2.60  | 67.55± 2.26  | 70.55± 1.93  | 71.58± 1.97  | 74.48± 1.81  |
| raw                  | 51.64± 2.83  | 54.80± 2.13  | 54.28± 2.20  | 54.68± 1.85  | 56.96± 2.22  |
| t-loss               | 60.18± 1.44  | 62.74± 1.62  | 65.12± 1.57  | 65.31± 1.55  | 67.76± 1.73  |
| tnc                  | 58.38± 2.46  | 63.65± 1.89  | 68.91± 1.64  | 70.05± 1.68  | 72.25± 1.99  |
| vae                  | 52.48± 0.74  | 53.03± 0.83  | 53.32± 0.85  | 53.57± 0.84  | 55.00± 1.29  |
| contrastive-perm-inv | 60.38± 2.70  | 60.63± 2.67  | 60.68± 2.68  | 62.30± 2.58  | 63.32± 2.54  |
| contrastive-mlp      | **67.05± 3.00** | 67.49± 2.56  | 70.00± 2.85  | 71.71± 2.50  | 72.79± 2.24  |

[2]The training data for the head consisted of 50 percent prompt curves sampled from the SM and 50 percent curves sampled from the CG.

Since this task required comparing the prompt curve to each of the candidate curves, we used a quadratic decision rule for the head. Let $h_0$ denote the encoder representation of the prompt (upsampled to 100 points prior to being fed into the encoder), and $h_i, i > 0$ denote the representations of the choices. The head linearly projected these vectors into a lower-dimensional space, and chose between the alternatives using a dot product in this space. That is, we fit a model of the form $p_i \propto e^{(wh_i, wh_0)}$ where $w$ is a linear projection from the encoder space to $\mathbb{R}^{32}$ and $p_i, i = 1, 2$ are the choice probabilities. All heads were trained on a cross-entropy loss using the Adam optimizer with a learning rate of .01. We report the accuracy on a collection of 400 held-out curves (200 per class). In this case, we see from Table 2 an improvement in accuracy of approximately 5 to 10 percent compared with the second best model. Interestingly, the rank ordering of the models also differs compared to the categorization task: here cpc and tnc both outperform t-loss, and the vae generally matches performance of the raw encoder baseline.

## 5.3 Freeform extrapolation

In this task, for a given function $y \in \mathbb{R}^{100}$, the model was presented with an initial portion $y_{1:80}$ and required to make a prediction $\hat{y} \in \mathbb{R}^{20}$ that extended it for a fixed sized window (length 20). Performance was measured by $Sim(\hat{y}, y_{80:100})$, for some choice $Sim$ of similarity function. It has been argued that, due to its high-dimensional and underconstrained nature, this task provides a more rigorous test of extrapolation than do discrete categorization tasks and that, in an empirical setting, it may provide finer-grained insights into peoples' inductive biases (DeLosh et al., 1997). However, it may be unreasonable to expect that an algorithm

---

[2]More explicitly, to generate the CG completion, we computed the likelihood of the prompt with respect to each of the CG kernels, and then took the posterior mean of the kernel that attained the highest likelihood, mimicking the procedure of Schulz et. al.

trained without *any* predictive experience can exhibit a reasonable ability to perform free-form extrapolation. Here, we tested the hypothesis that this capacity can arise from modest amounts of supervised (predictive) training based on categorization judgements among function representations acquired in a self-supervised manner from the contrastive learning mechanism described above. To test this, we implemented a simple form of curriculum learning.

In the first phase of the curriculum, we trained a logistic regressor on the encoder features learned during unsupervised training, to predict the generative kernel of an input function, exactly as in the categorization task from Section 5.1. Here we presented the regressors with 300 functions and corresponding category annotations from each kernel. In the second phase of the curriculum, we present a small number of functions $y^k$ with no label annotations. We then fit a simple class-dependent forecasting model of the form:

$$y_i^k \sim w_0^{\hat{c_k}} + w_0 + \sum_{j=1}^{L}(w_j^{\hat{c_k}} + w_j)y_{i-j}^k \tag{2}$$

where $\hat{c_k} \in \{1, 2, \ldots, 14\}$ denotes the kernel class of the function $y^k$ predicted by the logistic regressor. Here $L$ is a hyperparameter that controls the autoregressive time lag. We set $L = 20$ in all cases. The parameters $\{w_j^m\}_{0 \le j \le L, 1 \le m \le 14}$ are weights that are fit using least-squares. When given a function $y$ to extrapolate at test, we first estimated the class $\hat{c} \in \{1, 2, \ldots, 14\}$ of $y$ using the logistic regressor and encoder features. We then forecast it using the autoregression weights $\{w_j + w_j^{\hat{c}}\}_{0 \le j \le L}$.

We compared the results of this procedure with two controls. The first was an Ideal Observer model that was given access to the true underlying Gaussian kernel used to generate each function, information to which the other models were not privy. This model forecast a given function by computing the posterior mean with respect to the kernel on which it was trained. Since this model used Bayesian inference on the exact underlying distribution over functions, it represented the best performance that any model could attain. We refer to this as the "GPIO" (Gaussian process Ideal Observer) model. The second control was a simple autoregression model, that removed the categorization step in order to evaluate its contribution to the forecast quality. It used an unconditional forecasting model of the form $y_i^k \sim w_0 + \sum_{j=1}^{L} w_j y_{i-j}^k$ that ignored any category structure. The autoregression model was trained on exactly the same number of functions as the other forecasting models (with the number of curves used to train the logistic regressor included in this count).

We evaluated the extrapolation performance of each model using the Pearson correlation coefficient and L2 distance (see the Appendix for results of L2 distance) between the actual and predicted curves. We report the average values for 4200 held-out curves (300 per class). The results, shown in Table 3, are similar to those for the categorization task. All models substantially outperformed the autoregression baseline, indicating that even imperfect category information is helpful for extrapolation. The contrastive model performed better than any other model except the GPIO model. Several example extrapolations from the contrastive model are shown in Figure 3.

## 6 Comparison with human data

The multiple choice completion task from Section 5.2 was modeled after Experiment 1 in Schulz et al. (2017). An intriguing result of that experiment was that people were more likely to select the CG completion than the SM completion. We asked whether any of the models shares this property. To do this, we measured the difference in accuracy when the prompt curve was sampled from the CG compared to when it was sampled from the SM. More precisely, let $\{y_0^i\}_i$ denote a collection of prompt curves, $\{y_{CG}^i\}_i$ the corresponding completions generated by the CG, and $\{y_{SM}^i\}_i$ the completions generated by the SM. Furthermore, define $z_i$ to be a binary variable that indicates whether $y_0^i$ was sampled from the CG or from the SM. In our design, half of the prompt curves were sampled from the CG, meaning that $z_i$ assumes each of the two values with 50 percent probability. For a given model, the choice probabilities $\{(p_{CG}^i, p_{SM}^i)\}_i$ are given as in Section 5.2. Then the accuracy difference is defined by

$$\Delta_{acc} := \mathbb{E}_i(p_{CG}^i|z_i = CG) - \mathbb{E}_i(p_{SM}^i|z_i = SM) \tag{3}$$

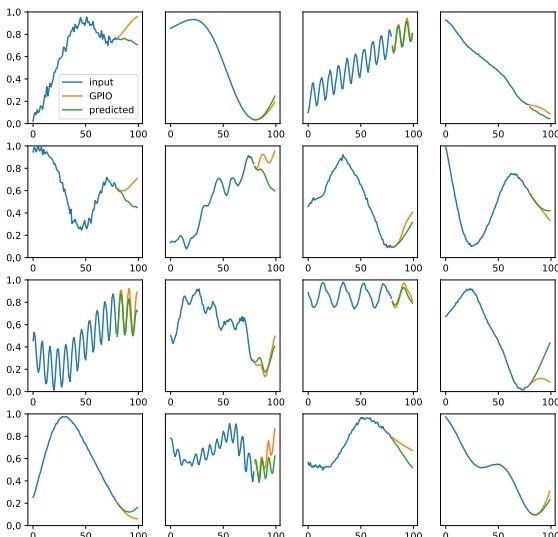

Figure 3: Freeform completions generated by the contrastive model, using the maximal amount of training data. The GPIO completions are also shown for comparison.

Table 3: Results on the freeform task, as a function of the number of samples per category used to train the regression (not counting the functions used to train the kernel classifier). Values are the Pearson correlation of the predicted to the true completion. The value for the GPIO model is 83.70± 1.78.

|                      | 1 | 3 | 10 | 30 | 100 |
|----------------------|---|---|----|----|-----|
| autoregression       | 18.48± 4.46 | 18.48± 4.48 | 18.47± 4.52 | 18.50± 4.47 | 18.66± 4.47 |
| contrastive          | **30.15± 2.36** | **49.05± 2.00** | **60.78± 1.59** | **63.34± 1.39** | **63.91± 1.43** |
| cnp                  | 24.29± 2.07 | 33.12± 1.98 | 40.91± 1.44 | 43.60± 1.33 | 45.37± 1.29 |
| cpc                  | 25.13± 2.83 | 42.32± 2.61 | 51.04± 2.39 | 53.06± 1.92 | 54.79± 1.92 |
| raw                  | 19.89± 4.20 | 23.13± 4.81 | 27.02± 5.82 | 28.18± 5.81 | 29.14± 5.89 |
| t-loss               | **29.45± 3.28** | 44.48± 2.74 | 54.62± 2.00 | 56.87± 1.93 | 58.10± 1.95 |
| tnc                  | **26.72± 3.22** | 39.11± 2.84 | 48.30± 1.81 | 51.52± 1.60 | 52.78± 1.74 |
| vae                  | **27.36± 3.09** | 33.51± 2.53 | 37.18± 2.67 | 39.27± 2.51 | 40.89± 2.39 |
| contrastive-perm-inv | 22.51± 2.65 | 32.33± 2.10 | 39.53± 1.52 | 42.57± 1.49 | 44.04± 1.33 |
| contrastive-mlp      | **26.72± 2.21** | 45.45± 2.41 | 56.19± 1.55 | 58.47± 1.65 | 59.64± 1.35 |

A short calculation shows that $\Delta_{acc}$ is directly related to the model's propensity to favor the CG completion over the SM completion:

$$\mathbb{E}_i(p^i_{CG}) = \frac{1}{2}\mathbb{E}_i(p^i_{CG}|z_i = CG) + \frac{1}{2}\mathbb{E}_i(p^i_{CG}|z_i = SM) \tag{4}$$

$$= \frac{1}{2}\mathbb{E}(p^i_{CG}|z_i = CG) + \frac{1}{2}(1 - \mathbb{E}_i(p^i_{SM}|z_i = SM)) \tag{5}$$

$$= \frac{1}{2} + \frac{1}{2}\Delta_{acc} \tag{6}$$

In other words, $\Delta_{acc}$ is positive exactly when the CG completion is chosen more often than the SM completion. Note that an unbiased model would have $\mathbb{E}_i(p^i_{CG}) = \mathbb{P}_i(z_i = CG) = \frac{1}{2}$ and thus $\Delta_{acc} = 0$. Note also that, as

Table 4: Value of $\Delta_{acc}$ on the multiple choice task, as a function of number of labeled training samples per category. The corresponding value for people is approximately 39. We do not bold the highest number because, unlike in the other tables, the values here do not correspond to a normative performance metric.

|  | 3 | 10 | 30 | 100 | 300 |
|---|---|---|---|---|---|
| contrastive | 19.82± 5.88 | 21.49± 5.26 | 20.53± 4.87 | 18.57± 4.84 | 19.39± 4.81 |
| cnp | 0.73± 5.39 | 5.25± 3.01 | 6.61± 4.35 | 8.17± 3.00 | 9.61± 3.32 |
| cpc | -3.92± 7.59 | 1.50± 5.50 | 4.73± 5.60 | 6.38± 4.43 | 10.33± 4.02 |
| raw | 5.19± 5.36 | 2.25± 3.79 | 3.87± 2.45 | 5.09± 1.81 | 7.92± 2.61 |
| t-loss | 0.57± 5.04 | 5.46± 4.37 | 8.24± 3.95 | 10.30± 3.44 | 12.72± 3.75 |
| tnc | -3.23± 6.13 | 4.70± 4.07 | 6.42± 2.81 | 8.18± 3.60 | 9.55± 3.97 |
| vae | 2.01± 1.34 | 2.62± 1.25 | 3.12± 1.42 | 3.30± 1.41 | 4.19± 1.79 |
| contrastive-perm-inv | 1.95± 5.01 | 4.60± 2.88 | 5.98± 4.05 | 8.17± 3.20 | 9.74± 3.55 |
| contrastive-mlp | 16.06± 3.74 | 16.56± 3.35 | 16.14± 4.01 | 16.48± 4.30 | 18.32± 3.84 |

described in Section 4, all models were trained on an equal proportion of curves from the SM and CG, so any resulting bias cannot be due to differential data availability between the two classes of curves.

We see from Table 4 that all models had at least a weak form of the CG bias, in that they attained higher accuracy on the multiple choice task when the prompt was sampled from the CG. The contrastive model had the highest value of this bias, albeit the error bars overlap with the values for t-loss and contrastive-mlp.

Crucially, however, even the baseline "raw" model showed a significant positive bias, thus indicating that the observed biases may be due to statistical properties of the curves themselves, independent of the properties of the learned representations of the models. The contrastive and contrastive-mlp were the only models that attained a $\Delta_{acc}$ value significantly higher than that of raw, thus indicating that the observed bias for these models is partially due to the properties of their representations.

However, all of these biases are smaller quantitatively than reported in Schulz et al. (2017). There it was found that people attain an accuracy of 32 percent when the prompt curve is from the SM (that is, they choose the CG completion 68 percent of the time), while they attain an accuracy of at least 71 percent when the prompt curve is from the CG. We say "at least", because in that experiment, there were actually three choices presented to the participants in the CG case: the CG completion, the SM completion, and an additional distractor completion. Thus we can estimate that, for people, the accuracy difference is given by

$$\Delta_{acc}^{people} \geq 39$$

# 7 Related Work

## 7.1 Contrastive Learning and Time Series Representation Learning

The idea of learning representations by maximizing an information theoretic criterion can be traced at least back to Linsker's InfoMAX (Linsker, 1988) principle, in which it was shown that certain properties of neurons in visual cortex could be replicated by training the encoder to maximize mutual information between the input and the encoder representation. This principle was subsequently extended to the problem of unsupervised deconvolution of time series (Bell and Sejnowski, 1995) by extraction of independent components. A network that learns by instead trying to maximize representational similarity between two different parts of the same input, presaging the modern approach to contrastive learning, was introduced by Becker and Hinton (1992). In a similar spirit, the BCM learning rule (Bienenstock et al., 1982), introduced as a model of synaptic plasticity in the visual cortex, can be shown to be equivalent to projecting the data onto subspaces that are "maximally discriminative" (Intrator and Cooper, 1992), and thus, most likely to be useful for downstream classification tasks. Rather than trying to optimize the mutual information directly, however, most modern implementations of this idea use a form of the InfoNCE loss, introduced by van den Oord et al. (2018). There

it was shown that this objective is a tractable lower bound to the mutual information criterion, which can be difficult to estimate directly. This loss is also strongly reminiscent of the older technique of Contrastive Hebbian learning (Hinton, 1989), insofar as both involve computing average network activations over a set of "positive pairs" of inputs as well as over a set of "negative pairs", and try to maximize the difference between the two averages. The authors incorporated this objective in their Contrastive Predictive Coding model (CPC), in which a recurrent encoder is trained to predict its own future outputs. This basic loss function has been adapted and modified in several ways. In Chen et al. (2020) it is used in tandem with a siamese network architecture, as we described in Section 2.1, while Aberdam et al. (2020) extends this setup to a seq-to-seq objective and Li et al. (2020) integrates the contrastive objective with a reconstructive one. A similar contrastive objective is used in He et al. (2020), except with a memory bank used to sample negative examples, with this approach extended to videos in Pan et al. (2021). The approach of Hyvarinen and Morioka (2016) is also very similar in spirit, in which the idea is to learn temporal features of time series that differ across different time windows.

Although some of the works above deal with sequential data, they tend to be high-dimensional (videos (Pan et al., 2021), image patches (Aberdam et al., 2020)) or data that is otherwise not directly interpretable by humans (audio (van den Oord et al., 2018), radio frequency signals (Li et al., 2020)), and do not consistently yield lower dimensional, readily interpretable, and easily composable functions of the form studied here. Fewer works have considered whether and how representation learning of such simple functions, such as the 1-dimensional time series of the sort used in function learning experiments and studied here-this omission is notable since, despite their simplicity, such functions occur in a wide range of naturalistic settings (Duvenaud et al., 2013) . Two particularly notable models of unsupervised time series learning are Triplet loss (Jean-Yves et al., 2019) and Temporal Neighborhood Coding (Tonekaboni et al., 2021). The first is inspired by word2vec (Mikolov et al., 2013), and relies on predicting the representation of a "word" (here a short window of the time series) from the representation of its "context" (here a longer window containing the "word"). In TNC, the timeseries is divided into disjoint segments. The encoder is jointly learned alongside a discriminator, in such a way that the discriminator is able to tell the difference between distant and proximal observations. In both TNC and Triplet loss, a recurrent encoder is used. An alternative approach to unsupervised learning of 1D time series learning is through autoencoders. A popular choice here is a seq2seq architecture with a reconstruction loss (Amiriparian et al., 2017; Lyu et al., 2018; Malhotra et al., 2017). Ma et al. (2019) augmented this setup with a k-means objective to encourage clustering in the latent space. Compared to our approach, these involve considerably more complexity, through the use of an additional decoding step, as well as more intricate seq2seq architectures.

## 7.2 Function Learning and Gaussian Processes

The dominant framework for modeling of human function learning uses Gaussian processes, a statistical model that specifies a probability distribution over the infinite-dimensional space of functions and allows for tractable inference procedures (Rasmussen and Williams, 2006). Lucas et al. (2015) used Gaussian processes to capture a wide range of empirical function learning phenomena, while Wilson et al. (2015) and Schulz et al. (2017) proposed specific families of kernels to model human extrapolation judgements. A limitation of the basic GP framework is its dependence on a choice of specific kernels or kernel families. Our approach sought to address this dependence through the use of unsupervised learning. Other approaches have taken a similar tack. The Spectral Mixture Kernel (Wilson and Adams, 2013) and Variational GP (Tran et al., 2016) do so by introducing nonparametric families of kernels that can approximate arbitrary kernels to an arbitrary level of precision. Duvenaud et al. (2013) implement a similar idea, except by building up a family of kernels using operations of a small number of atomic kernels, and performing a search over the resulting combinatorial space. Sun et al. (2018) perform a similar search except using a continuous relaxation and neural network. In Hinton and Salakhutdinov (2007), an appropriate kernel is found by fitting a Boltzmann machine. Neural Processes (Garnelo et al., 2018b) go further and replace the Gaussian kernel with a more flexible parametric family of distributions that can be learned using a neural network. This approach naturally extends to modeling of conditional distributions(Garnelo et al., 2018a; Kim et al., 2019; Gondal et al., 2021). This approach also has the advantage that the encoder can accommodate both variable number of observations, as well as variable x-locations. Such approaches share our broad goal of trying to learn the structure of a space of curves without assuming any particular functional forms ahead of time. However, both differ from ours in

that they build in more statistical machinery, by positing an explicit generative probabalistic model of the input distribution of curves.

## 8 Limitations

There are several limitations to our encoder. First, it differs from other models in that it was not designed to scale to very long time series. In particular, we use a feedforward convolutional encoder that processes the entire time series at once, while other techniques use some combination of recurrence and/or local windowing of the time series. Our time series have only 100 points, which is extremely short from the viewpoint of typical time series in learning models. However, in the context of function learning, such short time series have face validity, because during function learning experiments people can only make use of limited information at a time (Villagra et al., 2018). Thus, while it is not clear how well our encoder architecture would scale to very long time series, it is also not clear how well humans would do so either; and it remains to be determined how useful doing so would be for generalization in natural environments. These remain subjects for future research. Second, the feedforward nature of our encoder also restricts it to processing time series of a fixed length and sampling frequency, as opposed to the recurrent encoders of the other models which can handle time series of variable lengths, or the CNP-style permutation-invariant encoder, which can handle both variable lengths and variable sampling frequencies. In principle this could be overcome by upsampling or downsampling as necessary (and this was the approach we took in the multiple choice completion task). While this kind of resampling may be benign or helpful in certain circumstances (e.g., as a form of context normalization (Webb et al., 2020)), there are also many applications in which it would instead be preferable to preserve the original resolution and accommodate variable lengths. It is also possible that an appropriate modification of the CNP-style encoder could be used to overcome this difficulty; but due to the relatively poor results of the contrastive-perm-inv model, further work is necessary to fully flesh out this idea.

## 9 Discussion

The contrastive encoder we presented exhibited superior performance to comparison models in tests of generalization involving categorization as well as free form extrapolation. This was the case, despite its greater simplicity than those models. Moreover, we found that the performance was significantly degraded if an MLP was used in place of the convolutional layers in the encoder, and was even further degraded when using a Neural Process-style permutation-invariant architecture. On the other hand, the usage of a convolutional encoder is not sufficient on its own to achieve this level of performance, as shown by the poor results of the VAE, which employed such an encoder. This suggests that the combination of 1d convolutions, together with the contrastive loss and specific family of augmentations, may be key to learning good representations of intuitive functions.

In the multiple choice task, all models found it easier to correctly extrapolate prompt curves generated from the CG than curves generated from the SM, with this effect being most pronounced in the contrastive model. This is qualitatively similar to the corresponding empirical result from Schulz et al. (2017), regarding peoples' judgements in an analogous task. Thus our analysis suggests that such a bias may simply "fall out" as a consequence of a more general representation-learning procedure. More generally, we regard this as a proof of concept that the properties of representation learning algorithms can serve as an explanatory tool in the study of high-level human cognition such as function learning.

Indeed, several influential accounts of human intelligence posit the existence of elements of "core knowledge," such as an abstract number sense and fundamental notions of Euclidean geometry (Spelke and Kinzler, 2007; Chollet, 2019). Sometimes referred to as "atoms" of knowledge, these primitives are assumed to be low dimensional forms of representation and/or simple constructs and functions (e.g., continuity of processing, simple forms of causality), on which more complex cognitive abilities responsible for human intelligence are built. It has been proposed that the availability and use of such primitives is a critical factor in distinguishing human generalization capabilities from that of existing artificial systems (Lake et al., 2016), that rely on statistical estimation, and recent empirical evidence has been proposed in support of this claim (Kumar et al., 2020) Major efforts in cognitive science have assumed that such primitives are either genetically pre-specified,

or arise sufficiently early and predictably in development that they can be treated as predetermined. Based on this assumption, such efforts have focused research on the kinds of inference and learning mechanisms that, operating on such primitives, can compose them into more complex forms of processing (Lake et al., 2015; Ellis et al., 2020). Similarly, it has been proposed that such primitives should be considered as inductive biases when designing and comparing candidate computational architectures that seek to emulate human generalization capabilities. In contrast to this approach, some have argued that it is neither necessary nor accurate to assume that such primitives are pre-specific, but rather they arise from and are shaped by general purpose learning mechanisms interacting with and encoding statistical of present in the environment (Rumelhart and McClelland, 1986). While examples have been provided of how human-like concept formation and generalization can arise in this way (McClelland and Rogers, 2003), these have generally relied on externally supervised forms of learning that are explicitly trained on tasks that elicit such structure. To date, it has been difficult to design artificial systems that can discover low dimensional, simple forms of structure that can be exploited for generalization, using unsupervised or self-supervised forms of learning. Here, we have proposed one such mechanism, through a combination of contrastive learning and topological augmentations, and have demonstrated its ability learn to basic classes of functions, and simple compositions thereof.

For testing free form extrapolation, we used a curriculum learning strategy that involved first learning categories and then learning category-specific forecasting rules. While this procedure was more complex than for the other heads, there is reason to believe that it in fact resembles the process by which people may learn to make inferences in sparse and underdetermined settings. The most direct evidence of this in the realm of function learning comes from Leon-Villagra and Lucas (2019), which showed that peoples' extrapolations of curves were dependent on whether they judged the curves to lie in a previously encountered category, suggesting that people use category-dependent forecasting rules. More generally, our approach may be viewed as implementing a form of a Hierarchical Bayesian model, which have been show to capture the structure of peoples' intuitive theories about abstract structures in the world (Gershman and Niv, 2010; Tenenbaum et al., 2011; Kemp and Tenenbaum, 2008).

Our approach also fits with the idea that learning in natural agents involves adjudicating a tension between maintaining as much flexibility as possible (by optimizing a Maximum Entropy objective) while at the same time maximizing efficiency of computation (e.g., by optimizing a Minimum Energy objective). From this perspective, the contrastive encoder can be viewed as maximizing entropy, as implemented by the InfoNCE objective that we used, as it may be shown (Wang and Isola, 2020) that the second term in that objective is an estimator of the entropy of the distribution of codes in the latent space. Complementing this, our curriculum learning can be viewed as minimizing the energy of representations generated by a given category of function when presented with an instance of that function. This may strike a balance between flexibility (of generalization) and efficiency (of inference) that begins to approximate the balance observed in natural agents, and humans in particular (Frankland et al., 2021).

## 10 Broader Impact

As this work is concerned with foundational properties of learning algorithms in an abstract setting, we do not foresee any negative societal consequences arising directly from this work.

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
