# OpenReview forum: "A Self-Supervised Framework for Function Learning and Extrapolation"
_TMLR — Accepted by TMLR_

### Review · Reviewer_84zM · 2022-04-22

**Summary Of Contributions:**

This paper builds a framework for studying general representations learned to estimate "intuitive" scalar functions. These intuitive functions are constructed from a set of functions that have previously been shown to be more intuitive or easy for humans to learn, providing a reasonable proxy for naturalistic function learning. The paper uses a self-supervised encoder (e.g. an autoencoder) to learn representations which are invariant to topological distortions of these intuitive functions. The paper studies a few-shot learning paradigm where representations are pretrained on a large corpus of data, then a specialized final layer is learned few-shot mapping from representation to the desired predictor.

**Broader Impact Concerns:**

The included broader impact concerns section seems sufficient for this work.

**Requested Changes:**

* The changes marked "pedantic" above would be nice-to-have in order to improve readability.
* A deeper definition of Spectral Mixture functions and Compositional Grammar functions would significantly strengthen the readability.
* I think the work would benefit greatly from revisiting Table 4 and ensuring that it provides direct support for the claims the paper wishes to make. In order to better align with the abstract, I believe that a reinterpretation of Table 4 is necessary. However, the paper is strong even without this change.

**Strengths And Weaknesses:**

## Novelty, significance, relevance

The results and methodology in this paper appear novel. The work is certainly relevant to the representation learning community, though its connection to the psychology community is not immediately clear; however the psychology connection seems more of an aside in the paper anyways. The methodology represents a significant change to classical representation learning methodologies, where the experiment design is targeted towards emulating human behavior on a set of function learning tasks. The resulting experiments provide some interesting insights---though it's possible some of these insights may be misattributed (I will discuss more below).

## Correctness

Broadly, this paper did an exceptional job at matching empirical evidence to claims. Experiments were well designed with 30 samples used to estimate sample means of performance, uncertainty measures included in all tables, reasonable baselines provided, and special care given to training data to rule out potential sources of bias (i.e. data imbalance).

### Confidence intervals
95% confidence intervals are reported for every result in every table, however these confidence intervals are based on standard errors. It's not clear that proper assumptions are met for these 95% confidence intervals to be accurate, namely the normality assumptions on the underlying data as well as independence assumptions. There is underlying structure to the data (3 trials for 10 different hyperparameters = 30 total data points), which suggests that independence may be violated and also hints to me that normality plausibly would not hold. As a result, all of these uncertainty measures could be providing optimistically tight bounds on the sample mean estimate.

It should be noted, however, that in most cases the effect size appears to be quite large---there is a large difference in estimated mean performance between the proposed algorithm and baselines---so even if confidence intervals are optimistically tight, likely statistical significance still holds. These optimistic intervals may be adversely impacting conclusions in Table 3 where differences between the proposed method and the architecture ablation (contrastive-mlp) are often much smaller. In the other tables (Tables 1, 2, 4) these optimistic intervals likely do not impact conclusions severely.

### Table 4 -- bias towards CG

I believe the paper is actually underclaiming its results in this table. The bolding scheme and discussion in the paragraph at the bottom of page 9 suggest that the paper is seeking statistically significant differences between models. That is, the table is implicitly testing the claim:
> Does our proposed model (contrastive) induce greater bias towards CG functions than other models?

The resulting conclusion is that there are no statistically significant differences among the results. However, I believe the goal of this section should _actually_ be measuring difference from 0 bias, rather than differences between algorithms. That is, the claim should be:
> Do any of the models induce _any_ bias towards CG functions?

Answering this question would allow drawing conclusions that this empirical methodology yields similar biases as the human-centered methodology found in prior works. Interpreting the results in Table 4 in this way, it appears that the proposed method is statistically significant with a very meaningful difference (~20 points away from 0).

Regardless of which claim is measured, however, I wonder if there are confounding effects that are harming the conclusions. Specifically, the paper suggests that these biases towards better accuracy on CG functions vs SM functions are due to the modeling decisions themselves. To some extent, this is clearly true: there are notable differences in bias between models. However some degree of this bias very likely is coming from the data. Specifically, look at the "raw" baseline. The bias is approximately ~5 points in favor of CG, despite the model having no predictive power (it is effectively the identity function).

The differences in the underlying data could be due to the effective function spaces induced by CG vs. SM. Each function construction methodology yields a space of possibly observed functions. Imagine the extreme case that CG yields only one possible function, while the space for SM is extremely large. Then despite the data having 50% of examples drawn from each set, the singular CG function has 250k samples that the estimator can learn from, while each SM function has only 1 relevant sample. In this setting, then, it would be unsurprising that the estimator has some bias towards CG. Clearly the function space of CG is much larger than 1 function as in my example, but how large is the effective function space compared to SM? Considering this alongside the result from the "raw" baseline, does this suggest some of the differences found in the study are due to the underlying data instead of the learning mechanism?

As an aside, taking into account the possible underlying data bias and my suggestion to change the statistical test in Table 4, the proposed method actually looks _even better_. The proposed method (and the single ablation constrastive-mlp) would plausibly be the only methods that have a bias toward CG functions, suggesting that the learning mechanisms resemble human biases more strongly on this dataset than other learning mechanisms. (In order to draw this conclusion, I subtracted 5.19 [the raw bias] from each model's bias to approximate offsetting by underlying data structure. This isn't exactly correct, but might give a ballpark estimate of possible conclusions).

## Clarity

I had a couple of minor clarity concerns, primarily with the objective function stated in Section 2.1. The first, Section 2.1 notes InfoNCE in its title, however this is not actually discussed or even mentioned in the section body. A citation to a relevant paper is provided without context. If InfoNCE is important enough to be in the section title, likely it should be discussed in the section body.

The second clarity concern in Sec 2.1 is with the definition of the objective. The objective assumes that we are provided with pairs: $(v_i, v'_i)$ with $N$ such pairs. Later the objective is defined with modular arithmetic and a summation that spans from $i \in [1, 2N]$. It was not until reading the van den Oord et al. paper that I understood that the objective does not take advantage of the pairs structure (i.e. $v'$ is never used in the objective definition), but rather that these pairs are flattened into a single dataset of size $2N$.

I don't believe the Spectral Mixture functions are defined in this work---and even Compositional Grammar functions are only intuitively defined. I recognize both come from prior works, however they are critical to this paper and clarity would be greatly enhanced if a little more time was spent detailing these classes of functions.

## Pedantics

A few pedantic minor points that shouldn't influence the decision:
* There are several typos that shouldn't be possible with latex---notably several whitespace characters are dropped between a period and the start of the next sentence.
* On page 6, there is a paragraph segment (paragraph starts on page 5) which is nestled awkwardly between two figures. It looks like a caption and made reading awkward.
* Similar paragraph issue on page 8
* I had difficulty writing a summary of contributions based on the abstract and/or introduction. That is, neither the abstract nor introduction gave a clear overview of what is contributed by the paper. Notably, I found the abstract rather misleading in thinking this paper would somehow be related to reinforcement learning (terms like "agents" and "environments" and my own biases mixed poorly). This paper is highly specific to CG and SM functions and a particular methodology for constructing function learning. I believe the abstract should be more upfront about this, instead of being high-level and, well..., abstract.

---

> ### Author Response · Authors · 2022-06-01
> **Response to reviewer 84zM (pt 1)**
>
> Thank you for your detailed and helpful review.
>
> >This paper builds a framework for studying general representations learned to estimate "intuitive" scalar functions. These intuitive functions are constructed from a set of functions that have previously been shown to be more intuitive or easy for humans to learn, providing a reasonable proxy for naturalistic function learning. The paper uses a self-supervised encoder (e.g. an autoencoder) to learn representations which are invariant to topological distortions of these intuitive functions. The paper studies a few-shot learning paradigm where representations are pretrained on a large corpus of data, then a specialized final layer is learned few-shot mapping from representation to the desired predictor.
>
> Thank you for your detailed reading. There is however one detail in this summary which we believe may be a misunderstanding, which we will point out for the sake of maximal clarity: namely, our encoder is not an autoencoder since we do not apply a reconstruction objective. However, we do consider an autoencoder (more specifically, a VAE) as a comparison model.
>
>
> >95% confidence intervals are reported for every result in every table, however these confidence intervals are based on standard errors. It's not clear that proper assumptions are met for these 95% confidence intervals to be accurate, namely the normality assumptions on the underlying data as well as independence assumptions. There is underlying structure to the data (3 trials for 10 different hyperparameters = 30 total data points), which suggests that independence may be violated and also hints to me that normality plausibly would not hold. As a result, all of these uncertainty measures could be providing optimistically tight bounds on the sample mean estimate.
>
> We agree that these may be concerns-thank you for pointing them out. We consider the independence concern first. As you note, the observations have a 3x10 structure, corresponding to three copies of each model with different random weight initializations, evaluated on 10 hyperparameter combinations. It may well be that certain hyperparameter values are overall easier or harder than others for all models, thus leading to a violation of independence. (For example, suppose that we knew that certain hyperparameters theta1,...,theta9 were all very easy and that theta10 was very hard. If we are now given the first 27 of 30 accuracy values for a given model and see that they are all very high, then we can conclude that the remaining 3 values will likely be very low-which should not be possible if the 30 values were sampled independently).  However, it seems like a mild assumption to suppose that the 3 observations corresponding to different initializations will be independent, *after having conditioned on the hyperparameters*. Therefore, we consider a regression of the form :
> Accuracy=C(hyperparameters)+C(model kind)+C(labeled training set size)+noise
> Where the “C” denotes that each variable is treated as categorical (i.e., the regression is an ANOVA), and “noise” denotes iid Gaussians. Now, for each combination of (hparams, model kind, train size) we have 3 observations corresponding to different random initializations, which presumably are independent. The question then becomes whether the observed differences in model accuracies remain significant, after accounting for any main effects of different hyperparameters. We fit the above ANOVA separately for the classification, multiple choice, and freeform tasks. In all cases, we found that the 95 percent CI for the main effect of the contrastive model did not overlap with the CI for the next best performing model. We conclude that, even after accounting for potential dependencies in observations arising from hyperparameters, that the superior performance of the contrastive model remained significant.
>
> To address the normality concern, we recomputed the confidence intervals using a non-parametric bootstrap. For each set of 30 observations, we randomly sampled a collection of 30 observations (with replacement-so it could happen that some observations are repeatedly sampled) and computed the sample mean of these sampled observations. We repeated this 10000 times and then computed the bootstrap CI width as .5*((97.5 percentile sample mean)-(2.5 percentile sample mean)). In general, we found the values to be very close to those reported in the paper using Gaussian assumptions. Below we report the bootstrap CI intervals for the categorization (cf. table 1)

---

> > ### Comment · Reviewer_84zM · 2022-06-21
> > **Response**
> >
> > I appreciate the further statistical analysis here! It looks like results remain significant (which is unsurprising, given the strong effect sizes).
> >
> > I've read through all of the response comments above and looked through the update submission. I am quite happy with the modifications made to the paper.

---

> ### Author Response · Authors · 2022-06-01
> **Response to reviewer 84zM (pt2)**
>
>
> cnp                     &  1.257783 &  1.135759 &  0.733943 &  0.879182 &  0.945268 \\
>
> contrastive             &  2.034613 &  1.554182 &  1.270283 &  0.963125 &  0.877991 \\
>
> contrastive-cnp-encoder &  1.411339 &  1.370238 &  0.891667 &  0.653586 &  0.942277 \\
>
> contrastive-fc-encoder  &  1.583423 &  1.188095 &  0.923229 &  0.815491 &  0.677381 \\
>
> cpc                     &  1.520253 &  1.302396 &  1.119063 &  1.017887 &  1.126220 \\
>
> native                  &  2.478616 &  1.830402 &  0.951786 &  1.501786 &  1.764330 \\
>
> t-loss                  &  2.060818 &  0.866696 &  1.235134 &  1.070253 &  1.134583 \\
>
> tnc           &  1.500015 &  1.067857 &  0.875625 &  1.053571 &  1.281592 \\
>
> vae                     &  0.788705 &  1.350030 &  1.849479 &  1.736369 &  1.686339 \\
>
>
> And for multiple choice (cf. table 2):
>
>
>
> contrastive             &  2.287689 &  2.021435 &  1.983857 &  2.118521 &  1.922549 \\
>
> cnp                     &  2.293891 &  2.313796 &  2.091184 &  2.027018 &  2.292114 \\
>
> cpc                     &  3.301496 &  3.253497 &  3.002004 &  2.504115 &  2.480043 \\
>
> native                  &  2.102298 &  2.202733 &  1.720009 &  1.130312 &  2.086940 \\
>
> t-loss                  &  1.989409 &  2.091400 &  1.982490 &  2.141533 &  2.117908 \\
>
> tnc           &  3.248032 &  2.812632 &  2.960907 &  2.423094 &  2.514901 \\
>
> vae                     &  0.909834 &  0.834777 &  0.858265 &  0.874343 &  1.411075 \\
>
> contrastive-cnp-encoder &  2.631124 &  2.645153 &  2.672755 &  2.538904 &  2.497366 \\
>
> contrastive-fc-encoder  &  2.659382 &  2.601717 &  2.864080 &  2.460312 &  2.217216 \\
>
> Freeform (cf. table 3):
>
>
> contrastive             &  2.429204 &  2.242055 &  1.298574 &  1.396940 &  1.309118 \\
>
> cnp                     &  2.508449 &  2.204062 &  1.545111 &  1.268801 &  1.218795 \\
>
> cpc                     &  2.354891 &  2.681052 &  1.542335 &  1.501519 &  1.284492 \\
>
> native                  &  5.034393 &  5.069790 &  4.157291 &  3.944864 &  3.096929 \\
>
> t-loss                  &  2.783261 &  2.114933 &  1.183543 &  1.257299 &  1.167884 \\
>
> tnc           &  2.536687 &  2.351665 &  1.397038 &  1.260631 &  1.356301 \\
>
> vae                     &  3.165089 &  3.076054 &  2.528472 &  2.587125 &  2.379421 \\
>
> contrastive-cnp-encoder &  2.354410 &  1.876081 &  1.555415 &  1.434965 &  1.347386 \\
>
> contrastive-fc-encoder  &  2.132660 &  2.356822 &  1.645191 &  1.623197 &  1.358267 \\
>
> Deta_acc (cf. table 4)
>
>
> contrastive             &  4.825652 &  4.063974 &  4.069687 &  4.224868 &  3.135191 \\
>
> cnp                     &  5.348898 &  3.021717 &  4.311816 &  2.849094 &  3.319823 \\
>
> cpc                     &  4.107721 &  3.593597 &  4.102503 &  4.161995 &  4.176302 \\
>
> native                  &  4.164658 &  4.489881 &  3.322645 &  2.332254 &  4.042192 \\
>
> t-loss                  &  4.052797 &  4.261447 &  4.547958 &  3.532817 &  3.555841 \\
>
> tnc\_catwindow           &  4.419175 &  3.867181 &  4.204277 &  4.301037 &  4.715877 \\
>
> vae                     &  1.292831 &  1.366521 &  1.519034 &  1.540985 &  1.949353 \\
>
> contrastive-cnp-encoder &  4.984605 &  2.870053 &  3.963211 &  3.218210 &  3.640228 \\
>
> contrastive-fc-encoder  &  3.673728 &  3.480353 &  3.981995 &  4.382049 &  3.920659 \\
>
>
> >It should be noted, however, that in most cases the effect size appears to be quite large---there is a large difference in estimated mean performance between the proposed algorithm and baselines---so even if confidence intervals are optimistically tight, likely statistical significance still holds. These optimistic intervals may be adversely impacting conclusions in Table 3 where differences between the proposed method and the architecture ablation (contrastive-mlp) are often much smaller. In the other tables (Tables 1, 2, 4) these optimistic intervals likely do not impact conclusions severely.
>
> Please refer to the above response-in particular in Table 3 the confidence intervals obtained by bootstrapping are generally very close to those reported in the paper (and this holds true for the remaining tables as well, as you predict).
>
> >I believe the paper is actually underclaiming its results in this table. The bolding scheme and discussion in the paragraph at the bottom of page 9 suggest that the paper is seeking statistically significant differences between models. That is, the table is implicitly testing the claim:
> Does our proposed model (contrastive) induce greater bias towards CG functions than other models?
> The resulting conclusion is that there are no statistically significant differences among the results. However, I believe the goal of this section should actually be measuring difference from 0 bias, rather than differences between algorithms. That is, the claim should be:
> Do any of the models induce any bias towards CG functions?
> We agree that this is an equally important question, and intended to note this at the beginning of this section: “We asked whether any of the models shares this property [bias towards CG]”.

---

> ### Author Response · Authors · 2022-06-01
> **Response to reviewer 84zM (pt 3)**
>
> >Answering this question would allow drawing conclusions that this empirical methodology yields similar biases as the human-centered methodology found in prior works. Interpreting the results in Table 4 in this way, it appears that the proposed method is statistically significant with a very meaningful difference (~20 points away from 0).
>
> The lack of bolding in table 4 may have caused some confusion here. We have clarified this in the caption of table 4:”We do not bold the highest number because, unlike in the other tables, the values here do not correspond to a normative performance metric.”
>
> >Regardless of which claim is measured, however, I wonder if there are confounding effects that are harming the conclusions.
> Specifically, the paper suggests that these biases towards better accuracy on CG functions vs SM functions are due to the modeling decisions themselves. To some extent, this is clearly true: there are notable differences in bias between models. However some degree of this bias very likely is coming from the data. Specifically, look at the "raw" baseline. The bias is approximately ~5 points in favor of CG, despite the model having no predictive power (it is effectively the identity function).
> The differences in the underlying data could be due to the effective function spaces induced by CG vs. SM. Each function construction methodology yields a space of possibly observed functions. Imagine the extreme case that CG yields only one possible function, while the space for SM is extremely large. Then despite the data having 50% of examples drawn from each set, the singular CG function has 250k samples that the estimator can learn from, while each SM function has only 1 relevant sample. In this setting, then, it would be unsurprising that the estimator has some bias towards CG. Clearly the function space of CG is much larger than 1 function as in my example, but how large is the effective function space compared to SM? Considering this alongside the result from the "raw" baseline, does this suggest some of the differences found in the study are due to the underlying data instead of the learning mechanism?
>
> We agree with this point, and regret that we failed to acknowldedge that the result of the “raw” model is potentially informative. We have highlighted this in the text and discussed some of the implications. (Section 6: “Crucially, however, even the baseline ``raw" model showed a significant positive bias, thus indicating that the observed biases may be due to statistical properties of the curves themselves, independent of the properties of the learned representations of the models. The contrastive and contrastive-mlp were the only models that attained a $\Delta_{acc}$ value significantly higher than that of raw, thus indicating that the observed bias for these models is partially due to the properties of their representations.”)
> We also agree that the relative sizes of the function spaces likely plays a role as well, however the precise quantification of this can be challenging. For example, one could argue that the SM has a *smaller* function space than the CG. Indeed:the SM could also  be regarded as a special case of the CG, since the formula defining the former involves a summation and product of RBF and periodic kernels. Furthermore, the CG contains non-stationary kernels (namely, those that involve linear components) while the SM kernels are necessarily stationary.  So from this perspective, your argument could be turned on its head, and used to argue that there should be a bias towards the SM!
> That said, there are standard measures such as spectral entropy which can be used to quantify the size of function spaces corresponding to single kernels. We had originally considered including an analysis along these lines, but were not sure how to adapt it to the CG which is a mixture of several kernels, and also has hyperparameters. If there is an accepted way to extend the notion of spectral entropy (or a similar measure) to such a setting, we would be happy to include such an analysis.
>
> >As an aside, taking into account the possible underlying data bias and my suggestion to change the statistical test in Table 4, the proposed method actually looks even better. The proposed method (and the single ablation constrastive-mlp) would plausibly be the only methods that have a bias toward CG functions, suggesting that the learning mechanisms resemble human biases more strongly on this dataset than other learning mechanisms. (In order to draw this conclusion, I subtracted 5.19 [the raw bias] from each model's bias to approximate offsetting by underlying data structure. This isn't exactly correct, but might give a ballpark estimate of possible conclusions).
>
> Thank you for this observation-as mentioned in the above response, we have now noted in Section 6 that the contrastive and contrastive-mlp are the only models which attain a bias significantly higher than the “raw” baseline.

---

> ### Author Response · Authors · 2022-06-01
> **Response to reviewer 84zM (pt 4)**
>
> >I had a couple of minor clarity concerns, primarily with the objective function stated in Section 2.1. The first, Section 2.1 notes InfoNCE in its title, however this is not actually discussed or even mentioned in the section body. A citation to a relevant paper is provided without context. If InfoNCE is important enough to be in the section title, likely it should be discussed in the section body.
>
> We apologize for this editing oversight, and have removed the reference to InfoNCE from the section title.
>
> >The second clarity concern in Sec 2.1 is with the definition of the objective.
>
> We appreciate this being pointed out, which we believe may be a misunderstanding due to the use of the modular arithmetic in the formula.  The objective does crucially use the pair structure to define an “attraction” between the items in the pairs. This is visible in the first term of the formula, which refers to the paired elements v_i and v_i+N. However,  for the purposes of the second “repulsive” term, it is  correct that the pair structure is not used, and the dataset is flattened to size 2N. In the text, we have restated the definition without using modular arithmetic (which may have been needlessly confusing) and clarified these points as well.
>
> >I don't believe the Spectral Mixture functions are defined in this work---and even Compositional Grammar functions are only intuitively defined. I recognize both come from prior works, however they are critical to this paper and clarity would be greatly enhanced if a little more time was spent detailing these classes of functions.
>
> We have added definitions of both the CG and SM in section 2.2
>
>
> >There are several typos that shouldn't be possible with latex---notably several whitespace characters are dropped between a period and the start of the next sentence.
>
> We were able to locate and correct one such example of this  (“...structure particular to time series.Finally…” in Section 4). If you could point us to other specific examples we would be happy to correct them as well.
>
>
> >On page 6, there is a paragraph segment (paragraph starts on page 5) which is nestled awkwardly between two figures. It looks like a caption and made reading awkward. Similar paragraph issue on page 8
>
> We have fixed both of these
>
> >I had difficulty writing a summary of contributions based on the abstract and/or introduction. That is, neither the abstract nor introduction gave a clear overview of what is contributed by the paper. Notably, I found the abstract rather misleading in thinking this paper would somehow be related to reinforcement learning (terms like "agents" and "environments" and my own biases mixed poorly). This paper is highly specific to CG and SM functions and a particular methodology for constructing function learning. I believe the abstract should be more upfront about this, instead of being high-level and, well..., abstract.
>
> We have added the qualifier “in the domain of scalar function learning” in the abstract to make more clear the scope of the contributions.
> >The changes marked "pedantic" above would be nice-to-have in order to improve readability.
>
> We have addressed these-See above
> >A deeper definition of Spectral Mixture functions and Compositional Grammar functions would significantly strengthen the readability.
> We have added this (see above)
>
>
> >I think the work would benefit greatly from revisiting Table 4 and ensuring that it provides direct support for the claims the paper wishes to make. In order to better align with the abstract, I believe that a reinterpretation of Table 4 is necessary. However, the paper is strong even without this change.
>
> We have added further discussion regarding how much of the observed bias can be attributed to properties of data vs toperopties of the learned representations, in light of the bias of the “raw” model. Please also refer to the detailed discussion in response to reviewer 1 (the paragraph beginning “The results from section 6 is very interesting, but I believe it requires either some discussion/explanation…”)

---

### Review · Reviewer_ESBW · 2022-05-10

**Summary Of Contributions:**

This paper proposes a self-supervised learning method for learning a representation of 1-dimensional intuitive functions, which allows for better extrapolation. Specifically, the paper proposes 1) a 1D-convolutional encoder, 2) the use of contrastive loss, and 3) data augmentation via several topological distortions inspired by cognitive science literature. The experimental results show that the proposed method outperforms several baselines (e.g., CPC, triplet loss, VAE) on kernel classification, multiple choice extrapolation, and free-form extrapolation. In addition, the empirical result also shows that the proposed method has a bias towards Compositional Grammar (CG) like humans, which arguably makes it suitable for extrapolation in natural environments.

**Broader Impact Concerns:**

I do not have any ethical or broader impact concerns out of this paper.

**Requested Changes:**

1. Show how much each topological distortion improves extrapolation.
2. (Minor) Giving some examples of CG and SM kernels in the main text would be helpful for readers who are less familiar with them.
3. (Minor) Fix the wrong reference (Figure 3 -> Figure 1) in Page 4.
4. (Minor) Fix squeezed tick labels (texts) in Figure 1, 2, 3.

**Strengths And Weaknesses:**

* Strength
  1) The problem of learning a representation of 1D intuitive functions and extrapolating them is not only new but also very well-motivated by the cognitive science literature as discussed in the paper. While the majority of the prior work has focused on learning representations of images and texts, this paper introduces a relatively under-explored problem and makes a nice attempt to mimic how humans learn and extrapolate functions.
  2) The proposed way of applying topological distortions sounds plausible and convincing.
  3) The empirical result looks convincing. Although there is little work on learning representations for 1D functions, the proposed method is still compared against reasonable and popular representation learning algorithms including CPC, Triplet loss, and VAE.
  4) The paper is very well-written. The problem is very well-motivated, and the description of the method and experiment is very clear.

* Weakness
  1) There is no ablation study on the use and the specific choices of topological distortions. The proposed algorithm is a combination of 1) convolutional architecture, 2) contrastive loss, and 3) topological distortions. While there are ablation studies show the effectiveness of the first two, I could not find experiments showing the effectiveness of topological distortions unless I missed something. Showing this would make the result much more comprehensive.

---

> ### Author Response · Authors · 2022-06-01
> **Response to Reviewer ESBW**
>
> Thank you for your review and helpful suggestions.
> >Show how much each topological distortion improves extrapolation.
>
> Thank you for this suggestion. We have performed an ablation study for each combination of the three distortions, which is included in the appendix. We found all 3 augmentations to contribute positively to downstream performance. Please refer to our response to Reviewer 1 (the paragraph beginning “That said, we appreciate the value of providing a more detailed consideration of why… “) for further description and discussion.
>
> >(Minor) Giving some examples of CG and SM kernels in the main text would be helpful for readers who are less familiar with them.
>
> We have added more explicit definitions of the CG and the SM in section 2.2 (previously these had been in the appendix)
>
> >(Minor) Fix the wrong reference (Figure 3 -> Figure 1) in Page 4.
> >(Minor) Fix squeezed tick labels (texts) in Figure 1, 2, 3.
>
> Thank you-we have made both of the above fixes

---

> > ### Comment · Reviewer_ESBW · 2022-06-20
> > **Thank you for the response.**
> >
> > My apologies for the late response.
> > Thank you for reflecting all of my comments.
> > I have checked the ablation study that I requested from the appendix.
> > I do not have any major concern at this point, and the paper looks very solid to me.

---

### Review · Reviewer_83P8 · 2022-05-22

**Summary Of Contributions:**

The authors study representation learning in the space of 1D functions generated by various types of GP kernels.
They train 6 different types of encoders using either varying architectures or contrastive training losses.
Subsequently, they compare the learned representations in three different supervised tasks by learning simple predictions head on top of the learned representation, using only small amounts of data.

The three tasks are:
1. Classifying which type of kernel (out of 14 possible choices) a given function is most likely from
2. Selecting one of two possible choices for continuation of a given function
3. "Freeform" predicting the continuation of a given function by predicting auto-regression coefficients.

The 14 available kernels are selected from two different families of kernels: 13 generated from a compositional grammar ("CG", out of linear, rbf, periodic) and one spectral mixture kernel. The authors compare their results with biases found in humans, which exhibit a tendency to select CG kernels over SM kernels when choosing, and find a similar bias in their results.

**Broader Impact Concerns:**

No broder impact concerns.

**Requested Changes:**

Please refer for details to the review sections above.

As most critical recommended change, I would suggest giving the paper a clearer "take away" message. The authors hint at potential relevance to social/biological sciences, but these are kept at a too high level. Furthermore, the discussion of experimental results should also better highlight potential conclusions. For example (but not necessarily these), things like: Why is InfoNCE better than the baselines? What mechanism leads to the bias get introduced in section 6? What does this tell us about human cognition? etc..

On the methodological side, two points should be addressed before I can recommend acceptance:
* As the fixing hyperparameters for kernels instead of learning them is an unusual choice, which I believe could have quite strong influences on the qualitative results, it would be good to have some additional justification/discussion and, ideally, ablation studies (in the appendix is fine if space is scarce) showing that they don't change the qualitative results.
* The results from section 6 are very interesting, but require more discussion, or even better additional experiments, further explaining how the observed bias is introduced. It should also be confirmed that the bias is not yet present on the training data (which, I believe, would point to a bug).

Lastly, please have a look at the "Nitpicks" in the review section above. However, these aren't critical for my recommendation.


Furthermore, please improve the clarity of (experimental) descriptions. These should be small fixes.
* Last paragraph of section 5.1 (the test setup)
* How was L2 penalty used
* Section 5.2: How was the kernel from the CG family chosen?





**Strengths And Weaknesses:**

# Strengths

- Focus on a "simpler" (compared to large scale, e.g. visual, tasks), yet sufficiently complex and well defined domain can generate interesting results.
- The authors compare against a broad set of baselines
- The authors performed a sizeable number of experiments for various baseline/hyperparameter/task variations.
- Interesting results in section 6

# Weaknesses

I have grouped my feedback below into three categories, "Clarity", "Lack of clearly formulated learnings" and "Methodological questions".
I've also added a "Nitpick" section containing more subjective opinions/minor issues that I would like to point out, but do not expect the authors to explicitly address in the rebuttal (but I think would strengthen the paper).

## Lack of clearly formulated learnings/contributions

The most critical weakness of this paper for me is that the contribution is not clear. The authors state that they "propose a framework for addressing these challenges", meaning how to learn a good representation in a self-supervised fashion. However, this is a well-researched problem and the proposed framework is not novel. Neither the idea of unsupervised pre-training and learning small task-specific 'heads', nor the used loss function are new.

The authors further claim novelty on the investigated domain and the newly proposed augmentations. While the domain itself is not new, I am indeed not aware of deep-learning research on this domain for the specific question of self-supervised contrastive learning, although I am not an expert in this area. However, I believe a stronger justification for why this area is of interest would strengthen the paper. The authors make references to biological generalization, but these references are vague and possible insights gained are not clearly specified.

This connects to the experimental section, in which the results are interpreted as "method X performs better than method Y", but no deeper insights are formulated or discussed, for example about biological insights or reasons why some representation learning methods might perform better than others.

## Clarity

At a few places, especially in the experimental section, I was not able to exactly understand the description:

- Last paragraph before section 5.1: To me this paragraph reads as if each value in table 1 is the average/std over 30 values (3 copies * 10 hyperparameter choices). But does that mean each copy-hyperparameter pair was only evaluated on a *single* test-datapoint? Surely a larger test-set was used?
- Section 5.1: How is the L2 penalty applied to the encoder? I thought the encoder remains fixed for this task?
- Section 5.2: "we constructed the candidate completion curves by computing the posterior mean with respect either to the SM kernel, or to the CG kernel family". How was the kernel from the CG family chosen? For a CG prompt, was the "correct" kernel chosen or a random one? For a SM prompt, was a random or the best fitting CG kernel chosen?

## Methodological questions

- I am unsure about the choice to fix the hyperparameters for each kernel instead of fitting them to each given function. Because then the multiple choice task is not just between different kernel *types*, of which there are the described 14 (e.g. linear, rbf, SM), but actually between different *kernels* with different hyperparameters, which should make the choice much easier. (This also relates to my point in the 'clarity' section about how the corresponding CG kernels were chosen)
- The results from section 6 is very interesting, but I believe it requires either some discussion/explanation or, even better, additional experiments explaining how this bias is introduced. It would also be good to confirm that the bias does not exist on the training dataset and that the observed bias is induced by the generalization of the learned representation.

## Smaller points / Nitpicks

- "...that insights gained here are likely to shed light on the ability of biological and artificial agents to generalize more broad". This is a strong claim that would benefit from support from either citations or a more detailed argument.
- I am not sure about the intuitive explanation of the InfoNCE loss: If one approximates LSE as max(z_i), then this z_i won't be the most similar negative example, but it will be the positive example because the sum in the denominator is over *all* pairs, not just negative ones.
- I think section 2.2 would greatly benefit from some visual examples for some of the kernels used.
- I find it unsurprising that a CNN compares favourably to an MLP with the same number of parameters (CNNs have surprisingly few). A more interesting (IMO) comparison would have been to a larger MLP.
- “Thus our analysis suggests that such a bias may simply “fall out” as a consequence of a more general representation-learning procedure. More generally, we regard this as a proof of concept that the properties of representation learning algorithms can serve as an explanatory tool in the study of high-level human cognition such as function learning, thereby opening up many exciting directions for future work.” I believe this claim requires some more concrete examples as support.

---

> ### Author Response · Authors · 2022-06-01
> **Response to reviewer 83P8 (pt1)**
>
> Thank you for your useful and detailed feedback.
>
> >The most critical weakness of this paper for me is that the contribution is not clear. The authors state that they "propose a framework for addressing these challenges", meaning how to learn a good representation in a self-supervised fashion. However, this is a well-researched problem and the proposed framework is not novel. Neither the idea of unsupervised pre-training and learning small task-specific 'heads', nor the used loss function are new. The authors further claim novelty on the investigated domain and the newly proposed augmentations. While the domain itself is not new, I am indeed not aware of deep-learning research on this domain for the specific question of self-supervised contrastive learning, although I am not an expert in this area. However, I believe a stronger justification for why this area is of interest would strengthen the paper. The authors make references to biological generalization, but these references are vague and possible insights gained are not clearly specified.
>
> We appreciate this concern-and we agree with the assessment that the main contribution concerns the application of deep learning to extrapolation in the domain  of scalar function learning and the proposed augmentations. We certainly agree that the idea of pre-training/heads are not novel, and  explicitly note this in the introduction (“a standard self-supervised learning algorithm”, “other works have taken this general approach…”)
>
> That said, we see how the “we propose a framework…” sentence could be (mis)interpreted as laying out the core contributions of the paper. We have revised  this to read: “Here, we propose to address these challenges by adapting and extending the general framework of the field of self-supervised learning…” in order to more clearly delineate the scope of the contributions.
> Regarding the reference to “biological generalization,” our thinking  is simply that if we can identify a small set of general learning principles that work well in a relatively simple domain (such as scalar functions), those principles might be useful as the foundation for generating  hypotheses about the learning process of biological agents in other more complex domains. To provide a specific example of how this approach might yield useful insights- the contrastive loss can be regarded as adjudicating a tension between entropy maximization on the one hand and energy minimization on the other [Wang/Isola 2020]. In turn, the work of [Frankland et. al., 2021] uses this very same idea to explain a wide variety of phenomena in a domain very different from scalar function learning, namely human psychophysics. This example was described in the Discussion section.
>
> Finally, to further address the concern about “why this area is of interest” and to clarify the relation to “biological generalization,we have added a paragraph to the discussion section which significantly expands upon the relevance of our work to longstanding debates in cognitive psychology.(The paragraph beginning “Indeed, several influential accounts of human intelligence…”)
>
> >This connects to the experimental section, in which the results are interpreted as "method X performs better than method Y", but no deeper insights are formulated or discussed, for example about biological insights or reasons why some representation learning methods might perform better than others.
>
> Here again, given our focus on simple domains, we restricted our statements about more specific biological insights to relatively broader statements.  For example, in the Introduction we say: “We hypothesize that such distortions reflect commonly occurring structure in the world, that may in turn have been discovered by the brain, either through evolution or early development, and used as a basis for generalization.” That said,  we appreciate the value of providing a more detailed consideration of why some methods perform better than others. Accordingly, we now provide an additional analysis that quantifies the effect of each of the three augmentations on downstream performance of the contrastive model which we believe does provide a deeper understanding of the contrastive model. To summarize, we found the horizontal bending transformation (T2) to have the largest marginal contribution to downstream performance- this is noteworthy as, of the three transformations, it is the one  that maps most obviously onto the fundamental notion of  “tolerance space” in the original Topological Visual Processing theory [Chen,2005]. This convergence in turns bolsters our above-quoted hypothesis that the usefulness of the augmentations will be a function of the extent to which they reflect commonly occurring structures in the world-as, indeed, the most useful augmentation turned out to be the one that had previously been singled out for its ecological relevance. This new analysis is presented in the Appendix.

---

> > ### Comment · Reviewer_83P8 · 2022-06-20
> > **Thank you for the revisions**
> >
> > Dear Authors,
> >
> > Thank you for the revision. I'm replying here to all three of your responses.
> >
> > > That said, we see how the “we propose a framework…” sentence could be (mis)interpreted as laying out the core contributions of the paper. We have revised this to read: “Here, we propose to address these challenges by adapting and extending the general framework of the field of self-supervised learning…” in order to more clearly delineate the scope of the contributions. Regarding the reference to “biological generalization,” our thinking is simply that if we can identify a small set of general learning principles that work well in a relatively simple domain (such as scalar functions), those principles might be useful as the foundation for generating hypotheses about the learning process of biological agents in other more complex domains.
> >
> > Thank you for the clarification in the introduction!
> > Regarding the connection to 'biological generalization'. While it still seems rather vague to me, after going through the various comments and re-reading the paper, I believe that your contribution is valuable and a tighter connection with biology would be out of scope of the paper.
> >
> > > Thank you for this suggestion. We have repeated the analysis of the multiple choice task using optimized rather than fixed hyperparameters for each function.
> >
> > Thank you for running this additional experiment and including it! I find it very reassuring to see that the results (largely) hold up. I assume that `native` here is the same as `raw` in the other table? I did notice that `tnc` in the new experiment performs has a much stronger bias than before, especially in the low data regime. While I belive it does not change your interpretation of the results, if you have some idea why this might be the case, it would be useful to include in the discussion (in the appendix).
> >
> > > We have added further discussion about the extent to which the bias can be attributed to the learned representations as opposed to statistical properties of the dataset.
> >
> > Thank you!
> >
> > > However, we must confess to not having a good idea of just how much more efficiently the parameters are used in a Convnet vs an MLP. If you have a specific suggestion about how much larger of an MLP would be appropriate, we would be happy to rerun the analysis.
> >
> > I don't have a specific suggestion and I don't think this is a critical point. However, if you were to run additional experiments, I think the most relevant performance comparison (in my subjective opinion) would be to provide each architecture with as many parameters as it requires before the performance increase with additional parameters starts to level off (i.e. such that we're not parameter, but data constrained).
> >
> >
> > And thank you for all the other changes and fixes!

---

> ### Author Response · Authors · 2022-06-01
> **Response to Reviewer 83P8 (pt2)**
>
> >Last paragraph before section 5.1: To me this paragraph reads as if each value in table 1 is the average/std over 30 values (3 copies * 10 hyperparameter choices). But does that mean each copy-hyperparameter pair was only evaluated on a single test-datapoint? Surely a larger test-set was used?
>
> We have clarified this to indicate that each model is evaluated on a held-out test set of 200 curves per category.
>
> >Section 5.1: How is the L2 penalty applied to the encoder? I thought the encoder remains fixed for this task?
>
> We apologize for the typo here-we have clarified that the L2 penalty is in fact applied to the logistic regression weights, not to the encoder.
> >Section 5.2: "we constructed the candidate completion curves by computing the posterior mean with respect either to the SM kernel, or to the CG kernel family". How was the kernel from the CG family chosen? For a CG prompt, was the "correct" kernel chosen or a random one? For a SM prompt, was a random or the best fitting CG kernel chosen?
>
> In both cases, the chosen CG kernel was taken to be the one among all kernels in the CG that maximizes the likelihood of the prompt curve, following Schulz et. al. If the curve was in fact sampled from the CG, this procedure nearly always picks out the “correct” kernel. We have clarified this in section 5.2.
>
>
> >I am unsure about the choice to fix the hyperparameters for each kernel instead of fitting them to each given function. Because then the multiple choice task is not just between different kernel types, of which there are the described 14 (e.g. linear, rbf, SM), but actually between different kernels with different hyperparameters, which should make the choice much easier. (This also relates to my point in the 'clarity' section about how the corresponding CG kernels were chosen)
>
> Thank you for this suggestion. We have repeated the analysis of the multiple choice task using optimized rather than fixed hyperparameters for each function. The qualitative results were very similar, however all models performed worse than in the fixed hyperparameter setting, as would be expected. These results are now presented in the Appendix.
>
>
> >The results from section 6 is very interesting, but I believe it requires either some discussion/explanation or, even better, additional experiments explaining how this bias is introduced. It would also be good to confirm that the bias does not exist on the training dataset and that the observed bias is induced by the generalization of the learned representation.
>
> Thank you for this suggestion. We have added further discussion about the extent to which the bias can be attributed to the learned representations as opposed to statistical properties of the dataset. Specifically, as pointed out by another reviewer, the “raw” baseline (which simply copies the input) attained a nonzero bias, indicating that curves from the CG may be intrinsically simpler, independent of the properties of any particular learning algorithm. So, it is true that a  bias does exist in the training data, and that this could account for the performance of some of the models. But this cannot account for the full effect, because the contrastive (and contrastive-mlp ablation) models attain values significantly higher than the “raw” baseline.  Those were the only models for which this was true, suggesting that they are the only ones for which the observed bias is induced by the generalization of the learned representation. This supports our original hypothesis that explicit incorporation of invariances that commonly occur in the world (bending, rescaling,reflection) would allow for superior generalization (cf. the above discussion beginning “Here again, given our focus on simple domains…”)
>
> >"...that insights gained here are likely to shed light on the ability of biological and artificial agents to generalize more broad". This is a strong claim that would benefit from support from either citations or a more detailed argument.
>
> Please refer to the discussion above (the paragraph beginning “Regarding the reference to “biological generalization”...”)
>
>
> >I am not sure about the intuitive explanation of the InfoNCE loss: If one approximates LSE as max(z_i), then this z_i won't be the most similar negative example, but it will be the positive example because the sum in the denominator is over all pairs, not just negative ones.
>
> We have changed the wording here-we now simply note that LSE is a monotonically increasing function of each of its arguments separately
>
> >I think section 2.2 would greatly benefit from some visual examples for some of the kernels used.
>
> We did provide such a figure in the appendix (Figure 4, Appendix F).

---

> ### Author Response · Authors · 2022-06-01
> **Response to Reviewer 83P8 (pt3)**
>
> >I find it unsurprising that a CNN compares favourably to an MLP with the same number of parameters (CNNs have surprisingly few). A more interesting (IMO) comparison would have been to a larger MLP.
>
> We appreciate this concern-we opted to fix the number of parameters across different encoders in order to discipline the possible architectures and to make the comparison as “apples-to-apples” as possible. But as you point out, this still may not be completely fair because different architectures may use their parameters more or less “efficiently” than others. However, we must confess to not having a good idea of just how much more efficiently the parameters are used in a Convnet vs an MLP. If you have a specific suggestion about how much larger of an MLP would be appropriate, we would be happy to rerun the analysis.
>
>
> >“Thus our analysis suggests that such a bias may simply “fall out” as a consequence of a more general representation-learning procedure. More generally, we regard this as a proof of concept that the properties of representation learning algorithms can serve as an explanatory tool in the study of high-level human cognition such as function learning, thereby opening up many exciting directions for future work.” I believe this claim requires some more concrete examples as support.
>
> We have added a new paragraph to the Discussion section which addresses this point, (“Indeed, several influential accounts of human intelligence…”)
>
>
> >As most critical recommended change, I would suggest giving the paper a clearer "take away" message. The authors hint at potential relevance to social/biological sciences, but these are kept at a too high level. Furthermore, the discussion of experimental results should also better highlight potential conclusions. For example (but not necessarily these), things like: Why is InfoNCE better than the baselines? What mechanism leads to the bias get introduced in section 6? What does this tell us about human cognition? Etc..
>
> Please refer to the above discussion. In summary, we have modified a key sentence in the introduction (“we propose a framework…” -> ”we propose to address these challenges by adapting and extending the general framework of the field of self-supervised learning…”) in order to more clearly lay out which aspects are and are not novel to our work.Secondly, we have clarified in the response the relation of our work to social/biological sciences, and given a specific example (the paper of Frankland et. al.) that we believe does show how the abstract principles identified in our work can be ported over to broader psychological applications. Furthermore, we have deepened the discussion of experimental results in two ways: firstly, through a new ablation study of the effect of each of the three augmentations (Appendix), and secondly, through a discussion of the extent to which the CG bias can be attributed to the the statistics of the curves themselves, vs. to properties of the learning algorithms (Section 6). Finally, We have added a new paragraph in the discussion section which provides more context about the significance of our results in a psychological context.
>
>
> >As the fixing hyperparameters for kernels instead of learning them is an unusual choice, which I believe could have quite strong influences on the qualitative results, it would be good to have some additional justification/discussion and, ideally, ablation studies (in the appendix is fine if space is scarce) showing that they don't change the qualitative results.
>
> We have implemented this change and found little difference in qualitative results-see above discussion and appendix.
>
> >The results from section 6 are very interesting, but require more discussion, or even better additional experiments, further explaining how the observed bias is introduced. It should also be confirmed that the bias is not yet present on the training data (which, I believe, would point to a bug).
>
> We have added further discussion along these lines-see above discussion.
>
>
> >Furthermore, please improve the clarity of (experimental) descriptions. These should be small fixes.
> Last paragraph of section 5.1 (the test setup)
> How was L2 penalty used
> Section 5.2: How was the kernel from the CG family chosen?
>
> We have addressed all 3-see above

---

### Comment · Reviewer_83P8 · 2022-06-12
**Please re-upload pdf**

Dear Authors,

thank you for your detailed response. I'm currently going through your points and the updated paper.
However, I noticed that the new pdf does not allow marking/copying text (at least it doesn't work on my computer nor tablet).
If you can reproduce this on your side, could you please upload another version which allows working with the text?

Thank you very much!

---

> ### Author Response · Authors · 2022-06-13
> **Reuploaded pdf**
>
> Thank you for identifying this issue, and apologies for the inconvenience. We were able to reproduce this issue on our side. We have uploaded a new copy of both the main text and the supplement which, on our side at least, does allow for copying text. Please let us know if you continue to have issues with the document.

---

### Decision · Action_Editors · 2022-06-30

**Recommendation:** Accept as is

**Comment:**

This paper studies representation learning in the space of 1D functions. It generates functions from a set of kernels: 13 kernels are drawn from a family of compositional kernels (CG) and one spectral mixture kernel (SM). The paper optimizes a contrastive learning objective with data augmentation and explores several architectures for the encoder. It tests the learned representations on several downstream tasks, such as extrapolation and kernel classification.

One of they key results is that the contrastive learner with a 1D convolutional backbone exhibits a similar bias towards CG extrapolation compared to SM as humans do, albeit not quite as strong.

The reviewers all appreciated the focus on simpler functions, the quality of the writing, and the breadth of the experiments. The comparison with human data (section 6) is especially interesting. The main concerns were:

- A lack of clarity in terms of the specific contributions of the paper.
- Fixing kernel hyperparameters of each kernel rather than fitting them to data.
- More detailed discussion of the results in section 6.
- Ablations on the effects of topological distortions in relation to extrapolation.
- The validity of the reported confidence intervals.
- Various clarifications and definitions (e.g., mathematical definitions of CG and SM kernels).

The authors addressed all of these points during the rebuttal period and unanimously recommend acceptance. In my opinion, this paper outlines a very nice fundamental study, clearly and convincingly supports its claims, and would make a solid contribution to the TMLR community.